# Care needs of patients with chronic wounds for implementing a virtual care program: A qualitative study

Nasib Babaei[1,2], Vahid Zamanzadeh [3], Leila Valizadeh[4], Mojgan Lotfi [5*], Marziyeh Avazeh[6]

1 Department of Medical Surgical Nursing, Faculty of Nursing and Midwifery, Tabriz University of Medical Sciences, Tabriz, Iran, 2 Department of Nursing, School of Nursing and Midwifery, Ardabil University of Medical Sciences, Ardabil, Iran, 3 Department of Medical Surgical Nursing, School of Nursing and Midwifery, Shahid Beheshti University of Medical Sciences, Tehran, Iran, 4 Department of Pediatric Nursing, School of Nursing and Midwifery, Shahid Beheshti University of Medical Sciences, Tehran, Iran, 5 Department of Medical Surgical Nursing, Faculty of Nursing and Midwifery, Tabriz University of Medical Sciences, Tabriz, Iran, 6 Department of Pediatric Nursing, Faculty of Nursing and Midwifery, Tabriz University of Medical Sciences, Tabriz, Iran

* mojgan.lotfi@yahoo.com

## Abstract

### Introduction

Chronic and complex wounds are serious public health problems worldwide. Given the time-consuming nature of chronic wound healing and the need for long-term follow-up, a virtual care approach can effectively manage these patients. Identifying the care needs of patients with chronic wounds is key to successfully managing their care remotely. This study aimed to identify the care needs of patients with chronic wounds for implementing a virtual care program to manage this group of patients remotely.

### Methods

This descriptive qualitative study was conducted using a conventional content analysis approach in wound care clinics of East Azerbaijan Province (northwestern Iran). Data were collected through six focus group discussions with wound therapists and six semi-structured individual interviews with patients with chronic wounds. Participants were recruited using purposive sampling. The data were analyzed by MAXQDA 10 software.

### Results

After analyzing the data, the most important care needs of patients with chronic wounds for implementing a virtual care program were identified into three main

not publicly available. Relevant de-identified excerpts are included in the article.

**Funding:** This work was supported by the Research Deputy of Tabriz University of Medical Sciences The funders had no role in study design, data collection and analysis, decision to publish, or preparation of the manuscript.

**Competing interests:** The authors have declared that no competing interests exist.

categories, including the need for awareness-raising, needs related to health dimensions, and the need for specialized financial support (insurance).

## Conclusion

The findings of this study indicated that the successful implementation of a virtual care program for patients with chronic wounds requires addressing three core needs: enhancing patients' awareness regarding wound management, attending to their physical, emotional, and social health dimensions, and providing financial support through insurance coverage for wound care services. Addressing these needs can significantly improve the quality of care and therapeutic outcomes for patients in a virtual care setting.

### Introduction

Chronic wounds are a worldwide problem. They negatively affect health-related quality of life due to physical damage, required treatments, the chronic nature of the condition, and the potential for recurrence [1]. The prevalence of chronic wounds and the associated healthcare costs are significant and continue to rise. Specifically, in Australia, Europe, and the Scandinavian countries, the costs of managing and treating chronic wounds account for approximately 2% to 4% of the total healthcare budget [2]. Although chronic wounds account for a significant part of the health care budget, they are not even considered a significant clinical problem in many countries. However, healthcare decision-makers need to know the extent of the problem so that they may provide appropriate care plans [3].

The rise in the elderly population and prevalence of chronic diseases such as diabetes contribute to the increase in chronic wounds, resulting in more complex and demanding care needs [4,5]. Many patients experience infection, increased pain, delayed wound healing, and reduced health-related quality of life due to poor home management of chronic wounds [6]. Chronic wounds require specialized and ongoing care, which in many cases is not fully met due to a lack of resources, inadequate training, and poor service delivery. This can result in longer treatment times and increased complications [7,8].

Given the time-consuming nature of chronic wound healing and the need for long-term follow-up, a virtual care approach can effectively manage these patients [9]. Virtual care encompasses any remote interaction between patients and healthcare providers and among providers using any information technology. These synchronous or asynchronous interactions are established to provide healthcare services for patient management at any time and place using various methods and models [10]. In this method, with a patient-centered approach, it is possible to facilitate continuous communication between patients and specialists, which leads to improved patient satisfaction and health-related quality of life. Emerging evidence suggests that telecare in chronic wound management increases access to specialized services and improves treatment outcomes [11]. Individuals with chronic wounds who receive

remote wound care services have improved access to timely wound care services, ease of communication with their wound care professional, and self-empowerment to manage their wounds [12].

Treating chronic wounds through a telehealth environment represents a new approach to wound care [13]. Research has emphasized that in integrating advanced technologies such as telemedicine to improve treatment adherence and patient outcomes, the development of innovative nursing interventions must be tailored to patient needs [14,15]. Although the virtual care approach shifts the care of patients with chronic wounds from medical settings to remote home care, it is essential to identify and meet the specific care needs of these patients to ensure optimal home care [16]. Even though identifying the care needs of patients with chronic wounds is key to successfully managing the care of these patients remotely, in an extensive literature review, no study was found that examined the care needs of these patients when providing virtual care. Therefore, this study aimed to identify the care needs of patients with chronic wounds for implementing a virtual care program to manage this group of patients remotely.

## Materials and methods

### Study design

The present study is a descriptive-qualitative study using a conventional content analysis approach in 2023–2024. A descriptive qualitative study provides a voice for individuals who experience the phenomena of interest and has the potential to transform nursing and midwifery practice, as well as healthcare services in general. It can offer policy recommendations within the areas under investigation and influence the delivery of healthcare [17]. In the present study, this design was chosen because the aim was to thoroughly investigate the perceptions and experiences of wound therapists and patients with chronic wounds regarding their care needs in implementing virtual care. This topic is subjective, complex, and context-dependent in nature and can only be understood through qualitative research.

### Setting and participants

The present study was conducted in the wound care centers of East Azerbaijan province in northwest Iran. There are six private wound care centers in East Azerbaijan Province, all under the supervision of Tabriz University of Medical Sciences. In each wound care center, two or three wound therapists, all of whom are nurses, work part-time under the supervision of a physician who also manages the center. All these centers were selected to collect samples for this study. A purposive sampling method was used for sample selection, aiming to achieve maximum diversity in terms of work experience, age, educational level, and different organizational positions. Therefore, the researcher consulted the managers of the wound care centers to select the wound therapists and patients with chronic wounds who met the inclusion criteria of this study. Inclusion and exclusion criteria were specifically developed for this study to ensure that the selected sample aligned optimally with the research objectives and target population. The inclusion criteria for wound therapists were having at least two years of experience working in wound care centers, at least two years of experience providing health care services to patients with chronic wounds, a bachelor's degree or higher. The inclusion criteria for these individuals were age over 18 years, having a wound for four weeks or more, ability to answer questions and cooperate as necessary. The exclusion criteria for wound therapists were resignation from their position as wound therapists and for patients who have hearing, vision, and mental health problems.

### Data collection

Data collection was performed from 3 May 2023–27 January 2024. The researchers used both individual interviews and focus group discussions to encourage diverse views in the interactive setting of focus group discussions and to obtain an in-depth understanding of the participants' experiences through the interviews. Focus group discussions and individual interviews were conducted by the first author, who holds a PhD in nursing with experience in qualitative research, while field notes were recorded by another author, a full professor with both qualitative research and clinical experience. Both

researchers maintained continuous reflexivity and reflective journaling to minimize potential biases during data collection and analysis. The semi-structured interview guide was developed based on a literature review and expert input, and piloted before data collection to ensure clarity and appropriateness.

First, in order to understand the implementation requirements of the virtual care program from the perspective of the participants, the desired data were collected through focus group discussions according to a predetermined schedule using a semi-structured interview guide along with audio recording and field notes. Due to the busy schedules of wound care therapists and the difficulty of coordinating a common time for all participants, the sessions were initially conducted online via the social network WhatsApp in four 2-hour meetings. This approach allowed participants to submit their responses as voice messages, text messages, and images of handwritten answers within 48 hours, ensuring that all opinions were received without scheduling conflicts. Following a schedule, the participants were asked a question through WhatsApp before the meeting. Within 48 hours of receiving the question, the participants sent their answers to the questions through the WhatsApp group. The researcher completed the unclear or missing information by asking additional questions online. During the 2-hour meeting, the focus group discussed all the answers received and reached a final conclusion. The research team analyzed the comments and answers received from each session separately. Finally, to facilitate deeper discussion and finalize the findings, we held two face-to-face meetings, each lasting two hours, at an appointed time to discuss and exchange opinions on any questions and final decisions with eight of the same participants, along with one wound clinic manager and one medical informatics specialist. This mixed approach enabled the collection of comprehensive and rich data.

The interviews began with a general question: Describe your experiences working in a wound clinic, and if you have experience providing telehealth services to patients with chronic wounds? Next, the main questions were asked to collect in-depth information. (Annex 1)

To complete the data, the researcher conducted individual interviews with seven patients with chronic wounds, who were selected using purposive sampling. In this way, the researcher attended the wound healing centers and selected people who were aware of and knowledgeable about the phenomenon in question with the help of the wound therapists of those centers. In addition to being willing, they would be able to discuss the details of the phenomenon under study. Data was collected through in-depth semi-structured interviews. Before the interview, the purpose of the study was explained to the participants. The time of the interview was coordinated with the participants over the phone. For this purpose, after the focus group discussion sessions, the patients were interviewed over the phone in their homes. The interview process began with more general questions, as the interview progressed and the data were simultaneously analyzed, more detailed questions were asked, and the participants were guided to further explain their understanding of what they were saying. (Annex 2) The duration of the telephone interview varied between 45 and 65 minutes.

To enrich the data collected, exploratory questions such as why, how, who, etc., were raised during focus group discussions and individual interviews. In this study, data saturation was reached when no new or contradictory information was obtained from the group and individual interviews.

## Data analysis

The analysis of the data obtained from the groups' discussions and individual interviews was done by MAXQDA-10 software using the conventional content analysis approach based on the proposed method (Graneheim and Lundman, 2004) [18] as revised by Granheim, Lindgren, and Lundman (2017) [19]. This method was chosen due to the absence of a theoretical framework or prior research on the care needs of patients with chronic wounds in a virtual care setting. In this approach, the categories and their names are directly derived from the data inductively, and the analyst will not use specific theoretical perspectives or predetermined categories. The advantage of this content analysis method is that the knowledge created is based on the participants' views and derived from the information in the actual context of the study [19]. This feature allows for the extraction of patient-centered actionable insights, which can serve as a foundation for designing virtual care interventions tailored to their needs.

In order to conduct data analysis, the first author transcribed all individual interviews and focus group discussions verbatim within 24 hours after each session. Each transcript was then reviewed by another member of the research team, who listened to the original audio recordings while comparing them to the written text to identify and correct any possible discrepancies. In addition, transcribed versions of the interviews were provided to several participants for verification of the accuracy and completeness of the material, and minor corrections were made based on their feedback.

Then, all the transcribed interviews and field notes were reviewed several times. The initial coding was conducted by the first author and then reviewed independently by two other members of the research team (an experienced qualitative research professor and another professor experienced in nursing and qualitative research). After reading the text several times, the words, sentences, or paragraphs in the transcripts were considered semantic units. Based on the hidden concepts in them, they reached the level of abstraction and conceptualization and were assigned specific codes. Then, a coding tree including codes, subcategories, and main categories was developed. Disagreements among the coders were resolved through group discussions and, if necessary, with the guidance of the supervisor. Data saturation was achieved when continuous data analysis yielded no new or contradictory information from the individual and group interviews. Data were analyzed concurrently with collection to determine saturation. After several rounds of review and coding, data collection ceased when no new codes or categories emerged. The codes were compared regarding similarities and differences and categorized as more abstract categories with specific labels. With the repeated and continuous comparison of the classes with each other by the research team and careful and deep reflection, the content hidden in the data was introduced under the main categories. The research team engaged in ongoing discussions to refine the coding and categorization process for the data, ultimately achieving consensus on the appropriate categorization scheme.

## Data trustworthiness

The criteria provided by Guba and Lincoln were used to ensure the accuracy and robustness of the data [20]. These criteria include credibility, dependability, confirmability, and transferability. The credibility of the data was ensured through continuous engagement with the research topic, the participants, and the data, along with data source triangulation using individual interviews and focus group discussions, as well as member checking. Member checking was conducted not only to verify the interview transcripts but also to validate the codes and categories. In total, seven participants (four wound care therapists and three patients) reviewed the transcribed interviews and the extracted categories, and their feedback was incorporated into the final analysis. The dependability of the data was achieved through maintaining an audit trail. All stages of data collection, transcription, coding, and analysis were meticulously documented, and reflexive notes, a coding log, and research team meeting minutes were maintained. These records were made available for review by two independent qualitative researchers to verify the consistency and reliability of the analytical process.

In order to increase the confirmability, two observers and experts in the field of qualitative research reviewed and monitored the process. These experts, who were not involved in data collection, reviewed a portion of the raw data, the initial coding tree, and the final category structure to ensure that the findings were derived from the data and were not influenced by researcher bias. Finally, transferability was enhanced by using purposive sampling with maximum diversity. In this sampling, demographic and professional characteristics of participants, including age, sex, education level, wound type and duration, as well as therapists' work experience, were considered to cover a wide range of perspectives and increase the applicability of the results to similar contexts.

## Ethics

This study was approved by the Ethics Committee at Research Vice-chancellor of Tabriz University of Medical Sciences (IR.TBZMED.REC.1400.771). Before collecting data, the study's purpose, methodology, and the procedures and timing of the interviews were explained to the participants. The written informed consent forms were obtained from all participants. In addition, the participants were reminded that participation in the study was voluntary, and they had the right to decline

participation at any time without facing any negative consequences. For those with limited literacy, the consent form was read aloud by the researcher, and written consent was obtained after ensuring full understanding. Confidentiality of the data was maintained through coding and secure storage of information.

## Results

In the present study, 12 informed individuals participated in focus group discussion sessions and seven in individual interviews (Table 1).

The results of this study showed that the most important care needs of patients with chronic wounds for implementing a virtual care program could be divided into three main categories: the need for awareness-raising, needs related to health dimensions, and the need for specialized financial support (insurance) and nine subcategories as discussed below (Table 2).

To examine the influence of participant characteristics on responses, the results showed that the core care needs of patients were consistent across all subgroups. However, some minor differences in expressing experiences were observed. For instance, patients with higher education provided more detailed and specialized descriptions of wound management, and patients with longer wound duration shared more experiences regarding daily challenges. Thus, the quotes presented in the text are directly linked to each subcategory and reflect the overall agreement of participants with the extracted categories.

**Table 1. Demographic information of participants.**

| Wound Therapists (n = 12) | Patients (n = 7) |
|---|---|
| **Gender**<br> Male: 9<br> Female: 3 | Male: 4<br>Female: 3 |
| **Age/ year**<br> (Mean ± SD: 35.3 ± 8.1) | (Mean ± SD: 44.25 ± 19.0) |
| **Marital status**<br> Single: 1<br> Married: 11 | Single: 1<br>Married: 6 |
| **Level of education**<br> Bachelor of Science: 7<br> Master of Science: 2<br> Ph.D.: 2<br> Physician: 1 | Illiterate: 2<br>Diploma and below: 3<br>Academic: 2 |
| **The position of participants**<br> Wound Therapist: 10<br> Manager: 1<br> Medical informatics specialist: 1 | **Job**<br>Unemployed: 2<br>Full-time employee: 2<br>Part-time employee: 1<br>Other: 3 |
| **Work experience in domain of wound/ year**<br> 1-5: 4<br> 6-10: 3<br> 11-15: 4<br> 16-20: 1 | **Duration of the wound**<br>Less than six months: 1<br>One to Six months: 1<br>Over a year: 5 |
| | **The main caregiver**<br>Self: 3<br>Spouse: 1<br>Other people: 3 |
| | **How to pay medical expenses**<br>Free: 2<br>Medical service insurance: 2<br>Supplementary insurance: 3 |

**Table 2. Results.**

| Categories | Sub-categories |
|---|---|
| **The need for awareness-raising** | Lack of familiarity with wound management |
| | Need for knowledge about diet |
| | Need for knowledge about pain control |
| **Needs related to health dimensions** | The need to improve the physical health dimension |
| | The need to improve the emotional health dimension |
| | The need to improve the social health dimension |
| **Need for specialized financial support (insurance)** | Pressure due to high cost of wound treatment |
| | The high cost of advanced and modern dressings |

### The need for awareness-raising

Patients with chronic wounds have much need for education on self-care. In this regard, the patients stated in the interviews that they had numerous questions every day about their wounds and how to care for them, especially when their wounds were fresh, it was their first time facing such a problem, and they did not have access to experts in this field to obtain the necessary information. This category had three subcategories: lack of familiarity with wound management, need for knowledge about diet, and need for knowledge about pain control.

#### 1. Lack of familiarity with wound management

Lack of familiarity with wound management was identified as one of the care needs of patients with chronic wounds for the provision of virtual care. The wound therapists stated that patients and families are unfamiliar with wound symptoms, the treatment process, and how to manage and care for wounds. Their need for education and acquiring information leads them to search for content in unreliable sources such as the Internet, including Google or social networks. This incomplete or incorrect information can lead to choosing the wrong path of treatment and become a hindrance to the healing process:

*"All wounds can be treated if the patient or the patient's family are informed and if they refer promptly. Most of these wounds become chronic and do not respond to treatment because of ignorance or wrong referral to a specialist or physician who lacks knowledge in the field of wounds."* (Wound therapist)

#### 2. Need for knowledge about diet

According to the wound therapists, patients with chronic wounds lack sufficient knowledge about diet. As emphasized by the patients, they do not have sufficient knowledge about diet, which affects the wound healing process.

Therefore, patient education should be considered when providing virtual care. One patient participant stated the following:

*"I eat a lot of watery foods to help my wound heal faster, and chicken and fish are also effective in healing my wound. I do not eat anything else."* (Patient 5)

This was also noted in the focus group discussions:

*"Many patients with chronic wounds do not know what type of nutrients to consume based on their disease to avoid causing problems for their underlying disease and interfering with the natural healing process of the wound. Therefore, providing the necessary education on a diet appropriate to the individual's disease is necessary."* (Wound therapist)

### 3. Need for knowledge about pain control

The need for pain control knowledge when implementing a virtual care program for patients with chronic wounds was another issue regarding the need for awareness that most participants in this study frequently mentioned. They reported that most patients complain of severe pain at some point. Because they are not prescribed painkillers, they do not know what to do if they are in pain. One patient stated:

> *"Since I started treatment with a new specialist, my wound has improved a lot, but now it hurts badly. Because it was numb before, I did not feel anything. I told the physician, and he said that this is good news; it is getting better. However, this pain that I have is bothering me a lot, and I do not know what to do to relieve it. For example, last night, I was crying because of the pain I had. My medicine is for the infection; it does not do much to relieve the pain."* (Patient 2)

### Needs related to health dimensions

Patients with chronic wounds have multiple care needs related to different aspects of health. The wound, comorbidities, and advanced age cause many problems in different aspects of health for these patients. Families may meet a few of these needs, but wound therapists and physicians neglect most of them. The identified subcategories demonstrate the importance of this issue.

### 1. The need to improve the physical health dimension

Reduced mobility and physical activity are the most common physical problems that patients with chronic wounds experience to avoid putting pressure on the wound. According to patient reports in interviews, chronic wounds have a negative impact on the daily activities of these patients, with affected individuals unable to perform their personal tasks as they did before the disease. In this regard, one patient stated:

> *"Since my leg was wounded, I cannot do what I used to do. The wound has affected my activities. For example, I cannot go out now so that I do not have to stand and put pressure on the wound. I am not doing anything right now. If I did not have the wound, I could do my normal activities like shopping, etc."* (Patient 1)

In the interviews, the patients stated that in such situations, they need help to access information or a person who can guide them in dealing with such problems. Therefore, the wound therapist must provide the patient and their family with the necessary training for virtual care.

### 2. The need to improve the physiological health dimension

Nutritional problems, sleep problems, and experiencing excessive pain are among the important physiological health issues that patients with chronic wounds need to improve. Patients stated that these problems had a negative impact on their lifestyle. Hence, this issue requires special attention from wound care professionals and therapists when providing virtual care to these patients. Patient 3 stated:

> *"This wound has affected my sleep a lot. I am in so much pain that I cannot sleep. I do not have any painkillers to use in such a situation. If I am not in pain, I sleep well, but on nights when I am in pain, I sit in my bed and tell myself that it will get better on its own."* (Patient 3)

### 3. The need to improve the emotional health dimension

The need to improve the emotional health dimension is another need related to the health dimension identified in this study for remotely managing patients with chronic wounds. According to the participants, most of these patients feel

hopeless about the progress of their treatment due to the prolonged healing of the wound. They are afraid and worried about the possibility of amputation and infection in their wound. In this regard, one patient stated:

*"The first time we went to the physician with the test results for an examination, he said: 'Because the bones of your leg have turned black, it must be amputated.' I was terrified. I do not know what some physicians are thinking or whether they do not know they should not directly tell the patient that their leg must be amputated instead of motivating them. That is enough for a person to think they have lost everything. The motivation that the family gave was very effective."* (Patient 7)

Participants in group discussions discussed this issue as follows:

*"Most patients with chronic wounds are worried because they lack awareness. They say this is the first time they have encountered such issues. In these situations, the emotional dimension is an important aspect of how to support the patient and, more importantly, how the family can stay by their side so that they can continue their treatment."* (Wound therapist)

### 4. The need to improve the social health dimension

Another subcategory of needs related to the health dimensions of patients with chronic wounds is the need to improve the social health dimension. The participants noted that chronic wounds have a potentially adverse impact on the social life of people with chronic wounds. The important issues reported were loss of social connections and personal independence and the difficulty of going back and forth to wound healing centers. One patient stated:

*"Because of the wound I have, I do not go anywhere. I always stay at home. I do not have much contact with my relatives and friends, and I do not see them. The reason is that I do not want to walk on my wound, so it heals faster."* (Patient 2)

### Need for specialized financial support (insurance)

The interviews with patients and wound therapists revealed a common belief about the importance of financial support for patients with chronic wounds when providing virtual care. The wound healing process for these patients is time-consuming and may require long sessions and multiple visits to the wound therapist. Transportation, dressings, and other medical and treatment costs are unbearable for most of these patients. As a result, the need for specialized financial support should be one of the main priorities of the health system when providing virtual care to these patients. This category has two subcategories: pressure due to the high cost of wound treatment and the high cost of advanced and modern dressings.

### 1. Pressure due to high cost of wound treatment

Chronic wound management imposes high costs on these patients. Participants believed that as wound care services are not covered by insurance, the patient should pay the entire cost. Most patients cannot afford the costs, and some discontinue treatment at different stages of the treatment process due to financial problems. One patient stated in this regard:

*"By God, the costs are too high. On Saturday, when I went to see the physician for the dressing, it cost 49,000,000 IRR for the dressing, medicine, and doctor. Insurance does not cover this. This amount is too high for us, but we have no choice."* (Patient 4)

This was also discussed in the group meetings as follows:

*"Much expense is imposed on the family even if they want to use ordinary and simple dressings done for patients at home. The costs of home care centers are also high. The costs for a patient who may need to change dressings*

*several times a week are very burdensome, even with ordinary dressing. Unfortunately, insurance companies still do not cover home care costs at all. It might seem that they do, but the costs are not fully covered!"* (Wound therapist)

**2. The high cost of advanced and modern dressings**

In focus group discussions and patient interviews, participants stated that modern dressings can improve the healing of chronic wounds. However, due to financial difficulties, most patients and their families cannot afford advanced, modern wound treatment methods. Some patients even refuse to use these types of dressings for wound management, which has negative consequences for the patient:

*"After the treatment of these wounds begins, families sometimes refrain from buying good quality dressings due to their high cost and make do with a few simple ointments and dressings, which sometimes makes the wounds chronic. A simple wound has gotten deep and infected, increasing its problems."* (Wound therapist)

## Discussion

Based on the data analysis, three categories were identified as the care needs of patients with chronic wounds for implementing a virtual care program to manage patients at home: the need for awareness-raising, needs related to health dimensions, and the need for specialized financial support (insurance).

One of the findings of this study on the care needs of patients with chronic wounds was the need to raise awareness about wounds and their management when providing virtual care to these patients. These findings are largely in agreement with previous studies. Mahmoudi *et al.* (2020) reported that one of the important issues in chronic wound care is the knowledge gap among patients and the general population and the lack of educational materials designed for these patients [21]. In this regard, Kuhnke *et al.* (2019) stated in a study that patients with chronic wounds have limited knowledge about prevention, complications, and wound care. They identified solutions to improve patient and family education. They stated that the designed educational materials should include information on the signs and symptoms of wound infection, preventive measures, and wound care [22]. Our study also identified significant knowledge gaps among patients, underscoring the need for targeted educational materials to enhance their awareness of wound care and infection prevention. Also in accordance with the results of Kardiyudian et al. (2021), patients with chronic wounds ask various questions about the treatment and progress of their wound condition, which indicates their inadequate knowledge about wound and wound care [23].

The present study's findings showed that health dimensions are the primary need for caring for patients with chronic wounds when providing health care. Consistent with the results of our study, Kapp *et al.* (2018) conducted a qualitative study in Australia to examine the quality of life of people with chronic wounds who were self-medicating. The findings indicated that chronic wounds adversely impact the physical, emotional, and lifestyle domains of these patients. Based on these findings, the impact of chronic wounds in the physical domain included pain and suffering, physical inactivity, and exacerbation of other health problems; in the emotional domain, it included feelings of hopelessness about the progress of treatment, distrust of health care professionals, and concerns about wound infection [24].

In this regard, a qualitative study by Monaro *et al.* (2020) stated that chronic wounds lead to imbalances and subsequent adaptations in maintaining individual and social integrity. They reported sources of this imbalance as wound-related pain, sleep and mobility disturbances, nutritional problems, problems associated with aging, decreased ability to perform household tasks, loss of freedom and independence due to unscheduled care, and self-consciousness about wound odor [25]. Both studies emphasized the multifaceted effects of chronic wounds on reduced mobility and physical activity, emotional problems such as hopelessness and worry about infection or amputation, and social challenges such as decreased independence and worries about wound odor, all of which were also observed in our study.

What distinguishes our study is its focus on patient needs within the context of virtual care, unlike previous studies that primarily examined routine in-person care. We have identified patient needs in the field of virtual service provision that should be addressed when providing virtual care in order to provide more complete and tailored wound management care, and to reduce the physical, physiological, emotional, and social burden caused by wounds by making changes in the care of chronic wounds.

Another care need raised in this study was specialized financial support (insurance), which has been less addressed in previous studies. In this study, participants stated that home wound care services are not covered by insurance, and given the high cost of wound treatment and the high cost of modern dressings, some patients discontinue treatment at different stages of the treatment process due to financial difficulties. In this regard, the study of Kapp et al. (2018) showed that treating chronic wounds and receiving adjuvant treatments and specialized care for wound management are costly and result in financial pressure on the patients and their families [26]. In addition, Mahmoudi et al. (2020) stated that the complexity of insurance laws and regulations and reimbursement methods for wound healing products imposes a significant burden on patients. Delays in the approval or reimbursement of costs associated with advanced treatments and other unconventional dressings can exacerbate the disease and complicate or even preclude effective treatment. Furthermore, the lack of reimbursement for some treatments that prevent ulcers (e.g., compression stockings are not covered unless the patient has an ulcer) results in higher costs for at-risk patients [27]. While studies by Kapp (2018) and Mahmoudi (2020) have reported the financial burden of chronic wound care, we have emphasized the need to review insurance laws and reimbursement policies to facilitate virtual care and patient access to innovative treatments. Given the findings of this study and the financial challenges experienced by many patients with chronic wounds, it seems that government and insurance systems supporting virtual care programs could be beneficial. However, to validate these results and provide more precise policy recommendations, it is necessary to conduct larger and more empirical studies in the future.

## Study limitations

This study has limitations that should be considered when interpreting the results. First, the perspectives of patients' families, who are the primary secondary caregivers at home, were not examined. Second, focus group sessions were conducted virtually due to therapists' busy schedules; while this facilitated participation, it limited the ability to observe non-verbal cues and may have affected the richness of the data. Despite the limited number of patients in this study, purposive sampling with maximum variation was used to address this limitation, ensuring the inclusion of patients from diverse age groups, wound types, and care contexts. Fourth, there was potential for researcher bias, which was reduced by independent coding and participant review of categories. Finally, the relative weakness of evidence in the area of virtual wound care made comparisons and conclusions across studies difficult, which was partially offset by the use of relevant resources in health and telemedicine.

## Conclusions

This study highlights a new perspective in wound care that could lead to improved virtual management of patients with chronic wounds. The present study identified various aspects of the care needs of these patients when providing virtual care, including the need for patients to be familiar with wound management, needs associated with health dimensions, and the need for financial support, mainly coverage for wound care services. Addressing these identified needs can play a crucial role in the success of implementing virtual care programs for patients with chronic wounds. Therefore, it is suggested that wound healing clinic managers and wound therapists consider these findings when planning virtual care services.

## Implications

Based on the present study's findings, therapists and wound healing center administrators can use these results to provide high-quality, safe virtual healthcare services. In addition, to reduce the burden of chronic wounds and improve quality

of life, wound care center administrators can utilize the views of medical experts in wound care and improve the performance of wound therapists.

## Suggestions for further research

Future research should explore the care needs of patients with chronic wounds from their families' perspectives to enhance the effectiveness of virtual chronic wound management. Also, future studies with larger and more diverse samples are recommended to validate, refine, and expand the findings of this study. Finally, further research is needed to investigate the impact of implementing a virtual care program, considering the care needs identified in this study, the patient's health status, and the progress of wound healing during the program's implementation.

## Supporting information

**S1 File. Annex.**
(DOCX)

## Acknowledgments

The authors would like to express their gratitude to all the participants in the study.

## Author contributions

**Conceptualization:** Nasib Babaei.

**Data curation:** Nasib Babaei.

**Formal analysis:** Nasib Babaei, Mojgan Lotfi.

**Investigation:** Nasib Babaei, Mojgan Lotfi, Marziyeh Avazeh.

**Methodology:** Nasib Babaei, Vahid Zamanzadeh, Leila Valizadeh.

**Project administration:** Mojgan Lotfi.

**Resources:** Mojgan Lotfi.

**Supervision:** Mojgan Lotfi.

**Writing – original draft:** Nasib Babaei, Mojgan Lotfi.

**Writing – review & editing:** Nasib Babaei, Vahid Zamanzadeh, Leila Valizadeh, Marziyeh Avazeh.

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
