## [Decision Letter · Decision Letter 0]

8 Sep 2025

PONE-D-25-26146Care needs of patients with chronic wounds for implementing a virtual care program: A qualitative studyPLOS ONE

Dear Dr. Lotfi,

Thank you for submitting your manuscript to PLOS ONE. After careful consideration, we feel that it has merit but does not fully meet PLOS ONE’s publication criteria as it currently stands. Therefore, we invite you to submit a revised version of the manuscript that addresses the points raised during the review process.

We look forward to receiving your revised manuscript.

Kind regards,

Erfan Ghadirzadeh, MD

Academic Editor

PLOS ONE

Journal Requirements:

This work was supported by the Research Deputy of Tabriz University of Medical Sciences.

Reviewers' comments:

Reviewer's Responses to Questions

**Comments to the Author**

1. Is the manuscript technically sound, and do the data support the conclusions?

Reviewer #1: Yes

Reviewer #2: Partly

2. Has the statistical analysis been performed appropriately and rigorously?

Reviewer #1: N/A

Reviewer #2: N/A

3. Have the authors made all data underlying the findings in their manuscript fully available?

Reviewer #1: Yes

Reviewer #2: Yes

4. Is the manuscript presented in an intelligible fashion and written in standard English?

Reviewer #1: Yes

Reviewer #2: Yes

5. Review Comments to the Author

Reviewer #1: Yes this manuscript is technically sound. I am a bit confused about the way FGD were conducted. Yes, data underlying this manuscript were fully available. The manuscript is written in standard English. I have some underlying comments to the authors.

1. Abstract: Please add the number of FGDs and IDIs in the abstract on pages 31-32.

2. The Introduction is well written.

3. Line 100-102, please move ethical consideration to a separate paragraph at the end of the methodology section.

4. Settings and participants: Please add purposive sampling method was used for participant selection on line 109.

5. Page 6, line 115, is a repetition regarding informed consent. Please remove.

6. Page 7, I am not convinced how the FGDs were reported. Please follow the COREQ checklist to report your study. Please check. The original 2007 paper by Tong, Sainsbury, and Craig mentions that COREQ is a 32-item checklist for interviews and focus groups in qualitative health research.

7. Please move the interview guide page 8, lines 149-158, to the annex.

8. Page 8, line 165 is a repetition, please remove.

9. Page 9, line 171, please report the mean duration of the interviews.

10. Page 9, move lines 172-179 to the annex.

Reviewer #2: I would like to thank for the opportunity to review this manuscript. The topic is timely and highly relevant, as the management of chronic wounds represents a growing global health challenge, both in terms of patient quality of life and healthcare system costs. Exploring how virtual care programs can be tailored to meet patient needs is of particular importance, especially in contexts where access to specialized wound care is limited. More broadly, digitalization has become a common and shared strategy across public health programmes to address priority health needs and reduce the distance between communities and healthcare providers. By investigating the perspectives of both patients and wound therapists, this study provides valuable insights that can inform the design of patient-centered digital/tele-health interventions.

Please find here below my comments to your work.

Abstract

• Clearly state the study design (e.g., “descriptive qualitative study using conventional content analysis”).

• Specify data collection methods (focus groups and semi-structured interviews).

• The abstract conclusions are somewhat general. They should align more closely with the specific findings.

Introduction

• Provide a stronger rationale for using a qualitative approach and explain why content analysis was chosen.

• Clarify the conceptual/theoretical background that supports the research (paradigm, perspective, or framework).

• Ensure references are updated (include the most recent literature on chronic wounds and virtual care, beyond 2020).

Methods

• Study design: The rationale for choosing descriptive qualitative design is not given. State why this was the most appropriate.

• Researcher reflexivity: Information on the interviewers (credentials, training, gender, role, possible biases) is missing.

• Participant selection: Purposeful sampling is mentioned but not explained in detail; provide justification for the sample size (eg. why authors feel is sufficient before saturation). Please also report on refusals/dropouts (response rate).

• Inclusion/exclusion criteria: Listed, but it is unclear whether these were adapted from previous studies or developed specifically for this study. Please specify.

• Data collection procedures:

1. Clarify why FGDs were partly online (WhatsApp) and partly in person—justify the mixed approach.

2. State whether the interview guide was piloted or validated.

3. Report if transcripts were returned to participants for verification (member checking).

4. Provide details on how consistency was ensured when translating or transcribing interviews.

• Trustworthiness: The manuscript mentions credibility, dependability, confirmability, and transferability following Guba and Lincoln’s criteria, but the description is too general. Please expand with concrete examples:

1. Credibility: clarify whether member checking involved only transcript review or also validation of themes and indicate how many participants were engaged in this process.

2. Dependability: specify whether an audit trail was maintained and how the research process was documented for external review.

3. Confirmability: detail the role of the two qualitative experts (e.g., whether they reviewed raw data, coding tree, or final themes).

4. Transferability: explain which participant characteristics were prioritized to ensure diverse perspectives and how these support applicability of findings to other contexts.

• Ethical aspects: Ethics approval is reported, but it is unclear how informed consent was obtained for participants with possible literacy limitations. Clarify how confidentiality and voluntary participation were guaranteed.

Analysis / Results

• Analysis process:

1. Describe the coding process in more detail (how many coders, how disagreements were resolved, whether a coding tree was developed).

2. Clarify whether thematic saturation was achieved and how this was determined.

3. Strengthen the description of measures to ensure trustworthiness (triangulation, audit trails, member checking).

• Presentation of results:

1. Participant quotations are included but identifiers are inconsistent (e.g., “participant 5,” “patient No. 2”). Standardize.

2. Demographic table is cluttered; age is missing in participants patients. Please simplify presentation of participant characteristics.

3. Ensure the link between raw data (quotes) and themes is explicitly demonstrated.

• Influence of participant characteristics: Table 1 shows variation among the seven patient participants in terms of education and wound duration. The manuscript does not explain whether and how these differences influenced responses (e.g., patients with longer wound duration or higher education level may express different needs). Clarify whether patterns were observed across subgroups, or state explicitly if no differences emerged.

Discussion and Conclusions

• Some conclusions are broader than what the data justify (e.g., strong policy recommendations from a small sample). Tone down overgeneralizations.

• Clarify how findings align with or differ from prior literature and highlight the unique contribution of this study.

• Limitations: already listed but expand on the impact of (1) small patient sample, (2) use of WhatsApp FGDs, and (3) potential biases due to researcher role.

• Sample size: The patient sample (n=7) is relatively small. Authors should discuss the implications of this limited number, particularly regarding generalizability/transferability of findings and the potential underrepresentation of certain perspectives.

• Consequences of sample size: Recommend highlighting the need for further investigation with larger and more diverse patient populations to validate and expand upon the findings.

• Implications: discussion should distinguish between what findings directly support and what remains speculative.

• Suggestions for future research are appropriate but could be more specific (e.g., testing the identified needs in implementation trials).

6. PLOS authors have the option to publish the peer review history of their article (what does this mean?). If published, this will include your full peer review and any attached files.

Reviewer #1: **Yes: **Sayeeda Tarannum

Reviewer #2: No

---

## [Author Response · Author response to Decision Letter 1]

19 Nov 2025

23-Oct-2025

Dear Dr. Erfan Ghadirzadeh,

We highly appreciate your valuable feedback and meticulous review process. The provided recommendations improved the quality of the manuscript as well enriched the text. All the amendments proposed by the reviewers were fulfilled.

Hopefully, this article finds an opportunity for publishing in PLOS ONE.

Best Regards,

The corresponding author

Response to Editor Comments

Comment Response

1. Please ensure that your manuscript meets PLOS ONE's style requirements, including those for file naming. The PLOS ONE style templates can be found at https://journals.plos.org/plosone/s/file?id=wjVg/

PLOSOne_formatting_sample_main_body.pdf 1. We have carefully reviewed our manuscript and ensured that it fully complies with PLOS ONE’s style requirements.

This work was supported by the Research Deputy of Tabriz University of Medical Sciences.

Please include this amended Role of Funder statement in your cover letter; we will change the online submission form on your behalf. 2. The Role of Funder statement has been updated as follows:

“The funders had no role in study design, data collection and analysis, decision to publish, or preparation of the manuscript”.

3. Your ethics statement should only appear in the Methods section of your manuscript. If your ethics statement is written in any section besides the Methods, please delete it from any other section. 3. We have ensured that the ethics statement appears only in the Methods section of the manuscript, and it has been removed from any other section.

Please confirm at this time whether or not your submission contains all raw data required to replicate the results of your study. Authors must share the “minimal data set” for their submission. PLOS defines the minimal data set to consist of the data required to replicate all study findings reported in the article, as well as related metadata and methods (https://journals.plos.org/plosone/s/data-availability#loc-minimal-data-set-definition)...

If there are ethical or legal restrictions on sharing a de-identified data set, please explain them in detail (e.g., data contain potentially sensitive information, data are owned by a third-party organization, etc.) and who has imposed them (e.g., an ethics committee). Please also provide contact information for a data access committee, ethics committee, or other institutional body to which data requests may be sent. If data are owned by a third party, please indicate how others may request data access. 4. The data that support the findings of this study are the full interview transcripts, which contain potentially identifying and sensitive information. Due to participant confidentiality, these data are not publicly available. Relevant de-identified excerpts are included in the article.

5. If the reviewer comments include a recommendation to cite specific previously published works, please review and evaluate these publications to determine whether they are relevant and should be cited. There is no requirement to cite these works unless the editor has indicated otherwise. 5. No specific publications were recommended for citation by the reviewers.

Response to Reviewer 1

Reviewer comments Author Response

1. Abstract: Please add the number of FGDs and IDIs in the abstract on pages 31-32. Thank you for your valuable feedback, which has helped us enhance the clarity of our study.

1. The numbers of FGDs and IDIs have been added to the abstract (pages 31–32).

2. The Introduction is well written. 2. Thank you for your positive feedback on the Introduction.

3. Line 100-102, please move ethical consideration to a separate paragraph at the end of the methodology section. 3. The ethical considerations have been moved to a separate paragraph at the end of the methodology section.

4. Settings and participants: Please add purposive sampling method was used for participant selection on line 109. 4. We have added that a purposive sampling method was used for participant selection.

5. Page 6, line 115, is a repetition regarding informed consent. Please remove. 5. The repeated statement about informed consent has been removed.

6. Page 7, I am not convinced how the FGDs were reported. Please follow the COREQ checklist to report your study. Please check. The original 2007 paper by Tong, Sainsbury, and Craig mentions that COREQ is a 32-item checklist for interviews and focus groups in qualitative health research. 6. Thank you for the suggestion. We have revised the manuscript to ensure full compliance in accordance with the COREQ (32-item) checklist as recommended by Tong, Sainsbury, and Craig (2007). The reporting of the FGDs has been revised for clarity.

7. Please move the interview guide page 8, lines 149-158, to the annex. 7. The interview guide has been moved to the annex1.

8. Page 8, line 165 is a repetition, please remove. 8. The repeated content has been removed.

9. Page 9, line 171, please report the mean duration of the interviews. 9. We have now reported the mean duration of the interviews.

10. Page 9, move lines 172-179 to the annex. 10. Lines 172–179 on page 9 have been moved to the annex2

Response to Reviewer 2

Reviewer comments Author Response

Abstract

• Clearly state the study design (e.g., “descriptive qualitative study using conventional content analysis”). Thank you for your valuable feedback, which has helped us enhance the clarity of our study. The study design has been clearly stated in the abstract as a “descriptive qualitative study using conventional content analysis.”

Abstract

• Specify data collection methods (focus groups and semi-structured interviews). The data collection methods, including focus groups and semi-structured interviews, have been specified in the abstract.

Abstract

• The abstract conclusions are somewhat general. They should align more closely with the specific findings. The conclusions have been revised to better reflect the study’s specific findings.

Introduction

• Provide a stronger rationale for using a qualitative approach and explain why content analysis was chosen. We have strengthened the rationale for using a qualitative approach and explained why content analysis was chosen in the manuscript.

Introduction

• Clarify the conceptual/theoretical background that supports the research (paradigm, perspective, or framework). The conceptual and theoretical background, including the research paradigm and framework, has been clarified.

Introduction

• Ensure references are updated (include the most recent literature on chronic wounds and virtual care, beyond 2020). We have updated the Introduction with recent literature on chronic wounds and virtual care.

Methods

• Study design: The rationale for choosing descriptive qualitative design is not given. State why this was the most appropriate. The rationale for choosing a descriptive qualitative design has been added.

Methods

• Researcher reflexivity: Information on the interviewers (credentials, training, gender, role, possible biases) is missing. Thank you for your valuable comment. We have added details on the interviewers’ credentials, experience, and reflexivity measures in the Methods section under “Researcher Characteristics and Reflexivity”.

Methods

• Participant selection: Purposeful sampling is mentioned but not explained in detail; provide justification for the sample size (eg. why authors feel is sufficient before saturation). Please also report on refusals/dropouts (response rate). A detailed justification for purposive sampling has been added, including the rationale for the sample size based on data saturation. Information on the response rate has also been included.

Methods

• Inclusion/exclusion criteria: Listed, but it is unclear whether these were adapted from previous studies or developed specifically for this study. Please specify. The inclusion and exclusion criteria, developed specifically for this study to fit the research objectives and target population, have been clarified in the Methods section.

• Data collection procedures:

1. Clarify why FGDs were partly online (WhatsApp) and partly in person—justify the mixed approach. The use of a mixed approach has been justified in the Methods section to explain its relevance and feasibility for the study context.

• Data collection procedures:

2. State whether the interview guide was piloted or validated. It has been clarified whether the interview guide was piloted in the Methods section.

• Data collection procedures:

3. Report if transcripts were returned to participants for verification (member checking). It has been reported whether transcripts were returned to participants for verification (member checking) in the Methods section.

• Data collection procedures:

4. Provide details on how consistency was ensured when translating or transcribing interviews. We have expanded the “Data trustworthiness” section to include concrete details on how credibility, dependability, confirmability, and transferability were ensured.

• Ethical aspects: Ethics approval is reported, but it is unclear how informed consent was obtained for participants with possible literacy limitations. Clarify how confidentiality and voluntary participation were guaranteed. We have clarified how informed consent was obtained from participants with potential literacy limitations and detailed the measures taken to ensure confidentiality and voluntary participation in the Methods section.

Analysis / Results

• Analysis process:

1. Describe the coding process in more detail (how many coders, how disagreements were resolved, whether a coding tree was developed). The coding process has been described in more detail.

• Analysis process:

2. Clarify whether thematic saturation was achieved and how this was determined. It has been clarified how thematic saturation was determined and that it was achieved during the analysis process.

• Analysis process:

3. Strengthen the description of measures to ensure trustworthiness (triangulation, audit trails, member checking). The description of measures to ensure trustworthiness—including triangulation, audit trails, and member checking—has been strengthened in the Methods section.

• Presentation of results:

1. Participant quotations are included but identifiers are inconsistent (e.g., “participant 5,” “patient No. 2”). Standardize. Participant identifiers have been standardized throughout the results section for consistency.

• Presentation of results:

2. Demographic table is cluttered; age is missing in participants patients. Please simplify presentation of participant characteristics. The demographic table has been simplified, and participants’ ages have been added to improve clarity.

• Presentation of results:

3. Ensure the link between raw data (quotes) and themes is explicitly demonstrated.

• Influence of participant characteristics: Table 1 shows variation among the seven patient participants in terms of education and wound duration. The manuscript does not explain whether and how these differences influenced responses (e.g., patients with longer wound duration or higher education level may express different needs). Clarify whether patterns were observed across subgroups, or state explicitly if no differences emerged. The link between raw data (participant quotes) and themes has been clarified. We have also addressed the influence of participant characteristics, explaining whether patterns were observed across subgroups or stating explicitly if no differences emerged.

Discussion and Conclusions

• Some conclusions are broader than what the data justify (e.g., strong policy recommendations from a small sample). Tone down overgeneralizations. The discussion and conclusions have been revised to avoid overgeneralizations and ensure that recommendations are aligned with the data.

Discussion and Conclusions

• Clarify how findings align with or differ from prior literature and highlight the unique contribution of this study. We have clarified how our findings align with or differ from prior literature and highlighted the unique contributions of this study.

Discussion and Conclusions

• Limitations: already listed but expand on the impact of (1) small patient sample, (2) use of WhatsApp FGDs, and (3) potential biases due to researcher role. The limitations section has been expanded to discuss the impact of the small patient sample, the use of WhatsApp for FGDs, and potential biases related to the researcher’s role.

Discussion and Conclusions

• Sample size: The patient sample (n=7) is relatively small. Authors should discuss the implications of this limited number, particularly regarding generalizability/transferability of findings and the potential underrepresentation of certain perspectives The implications of the small patient sample have been discussed, including its impact on generalizability/transferability and the potential underrepresentation of certain perspectives.

Discussion and Conclusions

• Consequences of sample size: Recommend highlighting the need for further investigation with larger and more diverse patient populations to validate and expand upon the findings. We have highlighted the need for further research with larger and more diverse patient populations to validate and expand upon the findings.

Discussion and Conclusions

• Implications: discussion should distinguish between what findings directly support and what remains speculative. The discussion has been revised to clearly distinguish between findings directly supported by the data and those that are more speculative.

Discussion and Conclusions

• Suggestions for future research are appropriate but could be more specific (e.g., testing the identified needs in implementation trials). In the revised version, research suggestions are more specifically stated to clarify the context for further research.

---

## [Decision Letter · Decision Letter 1]

14 Dec 2025

Care needs of patients with chronic wounds for implementing a virtual care program: A qualitative study

PONE-D-25-26146R1

Dear Dr. Lotfi,

We’re pleased to inform you that your manuscript has been judged scientifically suitable for publication and will be formally accepted for publication once it meets all outstanding technical requirements.

Kind regards,

Erfan Ghadirzadeh, MD

Academic Editor

PLOS One

Additional Editor Comments (optional):

Reviewers' comments:

Reviewer's Responses to Questions

**Comments to the Author**

1. If the authors have adequately addressed your comments raised in a previous round of review and you feel that this manuscript is now acceptable for publication, you may indicate that here to bypass the “Comments to the Author” section, enter your conflict of interest statement in the “Confidential to Editor” section, and submit your "Accept" recommendation.

Reviewer #1: All comments have been addressed

2. Is the manuscript technically sound, and do the data support the conclusions?

Reviewer #1: Yes

3. Has the statistical analysis been performed appropriately and rigorously?

Reviewer #1: N/A

4. Have the authors made all data underlying the findings in their manuscript fully available?

Reviewer #1: Yes

5. Is the manuscript presented in an intelligible fashion and written in standard English?

Reviewer #1: Yes

6. Review Comments to the Author

Reviewer #1: (No Response)

7. PLOS authors have the option to publish the peer review history of their article (what does this mean?). If published, this will include your full peer review and any attached files.

Reviewer #1: **Yes: **Sayeeda Tarannum

---

## [Editor Report · Acceptance letter]

PONE-D-25-26146R1

PLOS One

Dear Dr. Lotfi,

I'm pleased to inform you that your manuscript has been deemed suitable for publication in PLOS One. Congratulations! Your manuscript is now being handed over to our production team.

Kind regards,

on behalf of

Dr. Erfan Ghadirzadeh

Academic Editor

PLOS One